# LEARNING DISENTANGLED REPRESENTATIONS FOR IMAGE TRANSLATION

## ABSTRACT

Recent approaches for unsupervised image translation are strongly reliant on generative adversarial training and architectural locality constraints. Despite their appealing results, it can be easily observed that the learned class and content representations are entangled which often hurts the translation performance. To this end, we propose OverLORD, for learning disentangled representations for the image class and attributes, utilizing latent optimization and carefully designed content and style bottlenecks. We further argue that the commonly used adversarial optimization can be decoupled from representation disentanglement and be applied at a later stage of the training to increase the perceptual quality of the generated images. Based on these principles, our model learns significantly more disentangled representations and achieves higher translation quality and greater output diversity than state-of-the-art methods.

## 1 INTRODUCTION

Learning disentangled representations from a set of observations is a fundamental problem in machine learning. Such representations can facilitate generalization to downstream discriminative and generative tasks as well as improving interpretability (Hsu et al., 2017), reasoning (van Steenkiste et al., 2019) and fairness (Creager et al., 2019). Recent advances have contributed to various tasks such as novel image synthesis (Zhu et al., 2018) and person re-identification (Eom & Ham, 2019). Image translation is an extensively researched task that benefits from disentanglement. Its goal is to generate an analogous image in a target domain (e.g. cats) given an input image in a source domain (e.g. dogs). Although this task is generally poorly specified, it is often satisfied under the assumption that images in different domains share common attributes (e.g. head pose) which can be transferred during translation - we name those *content*. In many cases, the class (domain) and common attributes do not uniquely specify the target image e.g. there are many dog breeds with the same head pose. This multi-modal translation motivates the specification of the particular class-specific attributes that we wish the target image to have - we name those *style*. The ability to transfer the content of a source image to a target class and style has been the objective of several methods e.g. MUNIT (Huang et al., 2018), FUNIT (Liu et al., 2019) and StartGAN-v2 (Choi et al., 2020). Unfortunately, we show that despite their visually pleasing results, the translated images still retain many class-specific attributes of the original image resulting in limited translation quality. For example, when translating dogs to wild animals, current methods are prone to transfer facial shapes which are unique to dogs and should not be transferred precisely to wild animals. As demonstrated in Fig. 1, our model transfers the semantic head pose more reliably.

In this work, we analyze the class-supervised setting and present a principled objective for disentangling image class and attributes. We explain why LORD, introduced by (Gabbay & Hoshen, 2020), cannot be applied for multi-modal translation. We then show that introducing an additional style representation overcomes this issue and propose a practical method for high-fidelity image translation by learning disentangled representations. Our method achieves this in two stages; i) Disentanglement: disentangled representation learning in a non-adversarial framework, leveraging latent optimization and well-motivated content and style bottlenecks. ii) Synthesis: the disentangled representations learned in the previous stage are used to "supervise" a synthesis network that generalizes to unseen images and classes. As synthesis network training is well-conditioned, we can effectively incorporate a GAN loss resulting in a high-fidelity image translation model. Our approach illustrates that adversarial optimization, which is typically used for domain translation,

| Content | Style | FUNIT | StarGAN-v2 | Ours |

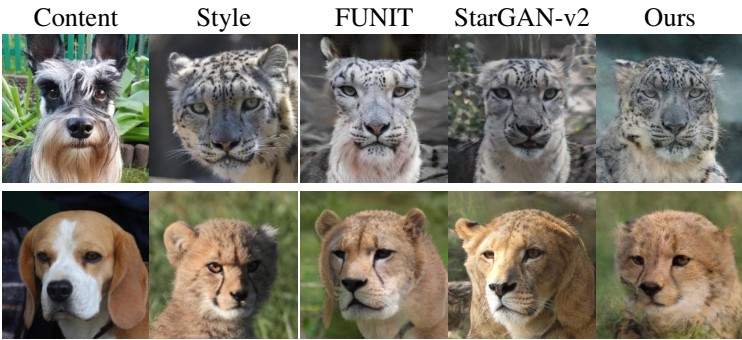

Figure 1: Examples of entanglement in state-of-the-art image translation models. Current approaches and their architectural biases tightly preserve the original structure and generate unreliable translations. Our model disentangles the high level content and captures the target style faithfully.

is not necessary for disentanglement, and its main utility lies in generating perceptually pleasing images. Our model learns disentangled representations and achieves better translation quality and output diversity than current methods.

Our contributions are: i) Introducing a non-adversarial disentanglement method that enables multi-modal solutions. ii) Learning statistically disentangled representations. iii) Extending domain translation methods to cases with many ($10k$) domains. iv) State-of-the-art results in image-translation.

### 1.1 RELATED WORK

**Image Translation** Translating the content of images across different domains has attracted much attention. In the unsupervised setting, CycleGAN (Zhu et al., 2017) introduces a cycle consistency loss to encourage translated images preserves the domain-invariant attributes (e.g. pose) of the source image. MUNIT (Huang et al., 2018) recognized that a given content image could be transferred to many different styles (e.g. colors and textures) in a target domain and extends UNIT (Huang & Belongie, 2017) to learn multi-modal mappings by learning style representations. DRIT (Lee et al., 2018) tackles the same setting using an adversarial constraint at the representation level. MSGAN (Mao et al., 2019) added a regularization term to prevent mode collapse. StarGAN-v2 (Choi et al., 2020) and DMIT (Yu et al., 2019) extend previous frameworks to translation across more than two domains. FUNIT (Liu et al., 2019) further allows translation to novel domains.

**Class-Supervised Disentanglement** In this parallel line of work, the goal is to anchor the semantics of all the images within each class into a separate class representation while modeling all the remaining class-independent attributes by a content representation. Several methods encourage disentanglement by adversarial constraints (Denton et al., 2017; Szabó et al., 2018; Mathieu et al., 2016) while other rely on cycle consistency (Harsh Jha et al., 2018) or group accumulation (Bouchacourt et al., 2018). LORD (Gabbay & Hoshen, 2020) takes a non-adversarial approach and trains a generative model while directly optimizing over class and content codes. Most works in this area demonstrate domain translation results on simple datasets but not in the multi-modal (many-to-many) settings. Moreover, their focus is to achieve disentanglement at the representation level rather than designing architectures for high-resolution image translation resulting in weak performance on competitive benchmarks. In this work, we draw inspiration from LORD in relying on the inductive bias conferred by latent optimization to learn a disentangled content representation. In contrast to LORD, we tackle the multi-modal image translation setting by modeling style. Moreover, we add an adversarial term in the synthesis network that increases the image quality and resolution.

## 2 BACKGROUND: REPRESENTATION LEARNING IN IMAGE-TRANSLATION

Image translation takes as input a set of $N$ images and corresponding class labels $(x_1, y_1), (x_2, y_2), ..., (x_N, y_N)$. Let us assume that an image $x_i$ is fully specified by its class $y_i$ and attributes $a_i$. As a motivational example, let us consider the images $x_i$ to be of animals, and the class label $y_i$ denotes the species. Attributes $a_i$, may include attributes $a_i^c$ common to all classes

such as head angle, and class-specific attributes $a_i^s$ such as dog or cat breed. An unknown function $G^*$ maps $y_i$, $a_i^c$ and $a_i^s$ to the image $x_i$:

$$x_i = G^*(y_i, a_i^s, a_i^c) \tag{1}$$

The goal of image translation is to replace the common attributes $a_i^c$ of target $x_i$ by $a_j^c$ of source $x_j$:

$$x_{ij} = G^*(y_i, a_i^s, a_j^c) \tag{2}$$

The main challenge is that during training, only the class $y_i$ of each image is given, but the attributes $a_i^c$ and $a_i^s$ are unknown. We define three representations corresponding to each of the physical properties above - the *class* embedding $e_{y_i}$ represents the class $y_i$, the *content* code $c_i$ represents the common attributes $a_i^c$ and the *style* code $s_i$ represents the class-specific attributes $a_i^s$. The computational task is to learn the representations $e_{y_i}, s_i, c_i$ such that they faithfully encode their corresponding physical properties of image $x_i$ and are sufficient to reconstruct the image using a generator $G$:

$$x_i = G(e_{y_i}, s_i, c_i) \tag{3}$$

This constraint however is insufficient to specify the representations uniquely. A popular constraint, utilizes the fact that common attributes $a_i^c$ are independent of the class $y_i$. In the motivating animals example, we assume that head pose acts similarly on cats or dogs. It therefore requires independence between the content code $c_i$ and class $y_i$.

*Adversarial methods:* Adversarial methods implement this constraint by training a domain confusion discriminator on the translated image $x_{ij} = G(e_{y_i}, s_i, c_j)$. The discriminator attempts to distinguish translated images from original images of the target class. Unfortunately, we empirically show that state-of-the-art methods do not learn content codes $c_i$ that are disentangled from the class $y_i$. The consequence is that $c_i$ transfers not only common attributes but also class-dependent attributes. We hypothesize that this failure is due to the challenging adversarial optimization.

In order to constrain the style $s_i$, current methods (e.g. StarGAN-v2, FUNIT) rely on locality-preserving architectures that bias the content code $c_i$ to represent the structure, while the style operates in a global manner per channel and is applied at a relatively deep level within the generator (typically from $16 \times 16$ spatial resolution). Unfortunately, such style is unable to model class-specific attributes $a_i^s$ of spatial nature e.g. the spatial facial shape of different dog breeds.

*Non-adversarial methods:* LORD is a non-adversarial approach for disentangled representation learning. It assumes that all attributes are common to all classes $I(a^c; y) = 0$ and $H(a^s) = 0$ - where $I$ denotes mutual information and $H$ denotes entropy. It therefore learns only class and content representations $e_{y_i}$ and $c_i$ but not style codes ($s_i = 0$). In order to learn disentangled class and content representations it uses a simple but effective constraint of minimal information in the content codes $c_i$. After learned disentangled representations $e_{y_i}$ and $c_i$, LORD learns a synthesis network which generalizes to unseen images and classes.

When the assumptions of LORD are satisfied ($H(a^s) = 0$), the content codes $c_i$ are indeed shown to recover the information of the common attributes $a_i^c$. However, when class-specific attributes exist, LORD does not achieve its goals. The reason is that the content code $c_i$, contains the information of both the common $a_i^c$ and class specific $a_i^s$ attributes ($I(a^s; c) > 0$). This has two issues: i) as the content code now contains class-specific information (e.g. cat breed), it is no longer disentangled from the class (we can predict if the content is of a cat); ii) when transferring content codes across classes (e.g. cats to dogs), we are missing the required class specific information in the target class (i.e. given the class code of $e_{dog}$ and content of a cat $c_{cat}$, the generator is not provided with the information of what dog breed to generate).

Due to its effective non-adversarial approach, LORD learns disentangled representations and achieves state-of-the-art results on uni-modal synthetic and low-res datasets. However, due to i) ignoring class-specific attributes as highlighted above; ii) using a non-adversarial *synthesis* network, it does not produce high fidelity images and cannot scale to high-resolution.

## 3 OVERLORD: PRINCIPLED DISENTANGLEMENT IN IMAGE TRANSLATION

Based on the above analysis, we present our disentangled image-translation method, OverLORD.

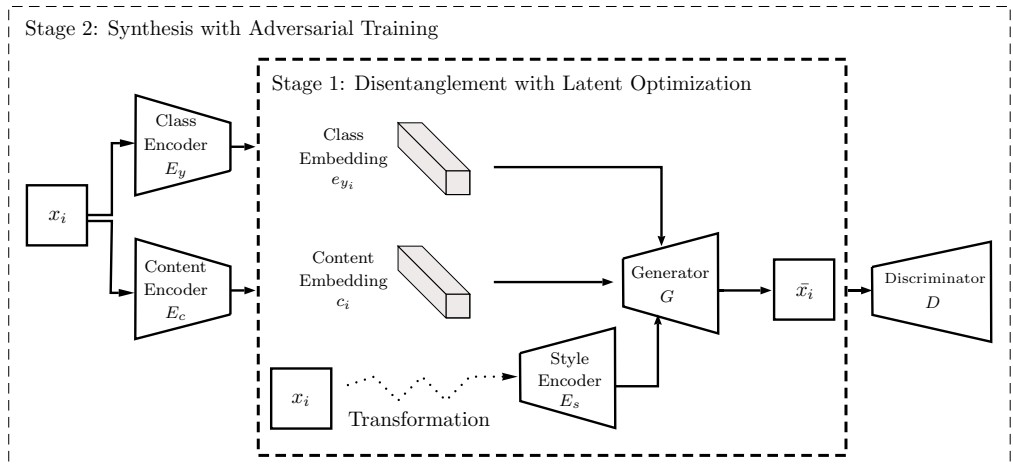

Figure 2: A sketch of the entire method. In the disentanglement stage, we optimize over a single class embedding per class and a single content embedding per image, as well as the parameters of the style encoder and the generator. In the synthesis stage, we optimize all modules in an amortized fashion, using the learned class and content embeddings as targets of the new encoders. During this stage, an additional adversarial discriminator is trained to increase the image fidelity.

### 3.1 LEARNING A DISENTANGLEMENT MODEL

In order to learn disentangled representations for class, content and style, we propose a non-adversarial framework with several principles.

*Reconstruction:* As the image should be fully specified by the representations $e_{y_i}, s_i, c_i$, we employ a reconstruction term: $\mathcal{L}_{rec} = \ell(G(e_{y_i}, s_i, c_i), x_i)$ - where $\ell$ is a measure of similarity between images (we use the VGG perceptual loss).

*Content bottleneck:* In order to constrain the amount of information in each content variable $c_i$, we parameterize it as a noisy channel consisting of a vector $c_i'$ and an additive Normal noise $z \sim \mathcal{N}(0, I)$, $c_i = c_i' + z$. Using the Shannon-Hartley theorem, the information capacity of the channel is a function of the signal-to-noise ratio. For the noisy channel $c_i$, the SNR ratio is given by $\|c_i'\|^2$. We therefore define the content bottleneck loss as $\mathcal{L}_{cb} = \sum_i \|c_i'\|^2$.

*Style:* We train a style encoder $E_s : \mathcal{X} \to \mathcal{S}$ to infer the class-specific attributes $a_s$. To encourage invariance to the common attributes $a_c$, the input image first undergoes a random transformation. This creates a transformed version $x_i^{trans}$ of the input image that shares the same class specific attributes $a_s$, but exhibits random common attributes $a_c$ (as style should be invariant to these attributes). The nature of the transformation is setting dependent e.g. if common attributes include spatial locations, then crop and rotation transformations will remove spatial attributes from the transformed image and make the style code $s_i$ invariant to them.

$$s_i = E_s(x_i^{trans}) \tag{4}$$

Our complete objective can be summarized as follows:

$$\min_{c_i', e_{y_i}, E_s, G} \mathcal{L}_{disent} = \sum_i \ell(G(e_{y_i}, E_s(x_i^{trans}), c_i' + z), x_i) + \lambda_{cb}\|c_i'\|^2 \quad z \sim \mathcal{N}(0, I) \tag{5}$$

There are several fundamental differences from the cVAE objective: i) the noise variance is a fixed hyper-parameter and is not learned; ii) there is no content encoder, instead the latent content code for each variable is learned directly, in an unamortized way often referred to as latent optimization; iii) the input to the style encoder is a transformed version of the image. Gabbay & Hoshen (2020) discovered that latent optimization improves disentanglement significantly over encoder-based methods. The difference lies in the initialization as latent optimization starts with no class-content correlation (code initialization is IID) while a content encoder starts with near perfect correlation (class can be predicted from the output of a randomly initialized content encoder). We further elaborate on latent optimization and its inductive bias in Appendix A.3.

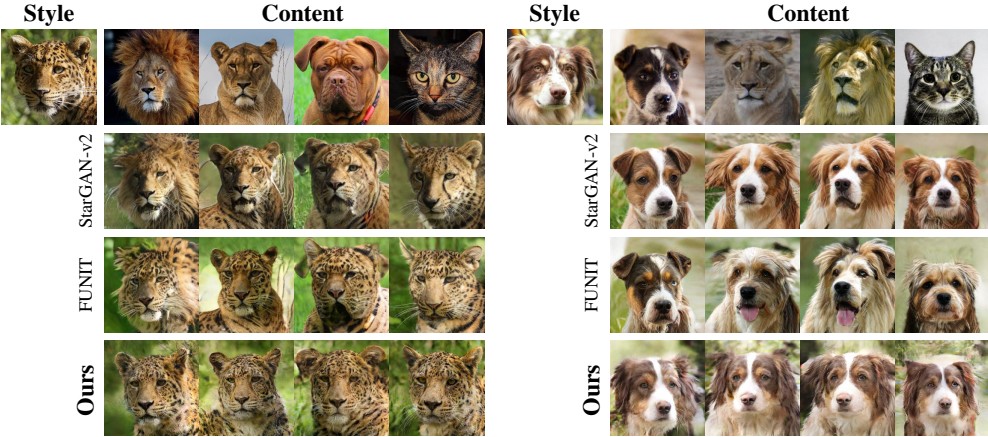

Figure 3: Comparison against best baselines on AFHQ (zoom in for better resolution). Class (i.e. cat, dog or wild) and style (e.g. breed) are guided by the reference image, while the content (e.g. head pose) of the source image should be preserved. StarGAN-v2 and FUNIT leak facial shapes that are unique to the source breeds leading to unreliable and inconsistent translations. Our method produces more disentangled results and captures the target style faithfully.

## 3.2 Generalization and Perceptual Quality

After the disentanglement stage, we possess a content code $c_i$ and shared class embedding $e_{y_i}$ for every training image, and also a trained style encoder $E_s$. In order to generalize to unseen images and novel classes, we follow LORD (Gabbay & Hoshen, 2020) and train a content encoder $E_c : \mathcal{X} \to \mathcal{C}$ and a class encoder $E_y : \mathcal{X} \to \mathcal{Y}$ to directly obtain a content code $c \in \mathcal{C}$ and a class code $y \in \mathcal{Y}$:

$$\mathcal{L}_{gen} = \sum_i \ell(G(E_y(x_i), E_s(x_i), E_c(x_i)), x_i) \quad \mathcal{L}_{enc} = \sum_i \|E_y(x_i) - e_{y_i}\|^2 + \|E_c(x_i) - c_i\|^2 \quad (6)$$

By optimizing the two terms, we ensure that the training set can be reconstructed in an amortized fashion ($L_{gen}$), without violating the disentanglement criteria established in the first stage ($\mathcal{L}_{enc}$). Note that the targets for $\mathcal{L}_{enc}$ are the ones learned by our own model in the first stage.

To increase the visual quality, we jointly train a discriminator $D$ and employ an adversarial loss:

$$\mathcal{L}_{adv} = \sum_i \log D(x_i) + \log(1 - D(G(E_y(x_i), E_s(x_i), E_c(x_i)))) \quad (7)$$

Our objective of the second stage can be summarized as follows (the style encoder is unchanged):

$$\min_{E_y, E_c, G} \max_D \mathcal{L}_{gen} + \lambda_{enc}\mathcal{L}_{enc} + \lambda_{adv}\mathcal{L}_{adv} \quad (8)$$

A sketch of the entire method is shown in Fig. 2.

## 4 Experiments

Our method is evaluated against state-of-the-art methods for multi-modal domain translation and class-supervised disentanglement. Note that we do not present comparisons to unsupervised disentanglement methods such as betaVAE as they are not competitive on class-content disentanglement with methods that use class-supervision. Implementation details are provided in Appendix A.1.

Table 1: Multi-modal domain translation on the three domains of AFHQ. Metrics: classification accuracy of class labels from content codes ($y \leftarrow c$), classification of original class from the translated image ($y_j \leftarrow x_{ij}$), translation fidelity (FID) and translation diversity (LPIPS).

| | Representation | Image | | |
|---|---|---|---|---|
| | $y \leftarrow c \downarrow$ | $y_j \leftarrow x_{ij} \downarrow$ | FID $\downarrow$ | LPIPS $\uparrow$ |
| MUNIT | 1.0 | 1.0 | 223.9 | 0.199 |
| DRIT | 1.0 | 1.0 | 114.8 | 0.156 |
| MSGAN | 1.0 | 1.0 | 69.8 | 0.375 |
| LORD | 0.74 | 0.47 | 97.1 | 0 |
| LORD w/ style clustering | 0.53 | 0.43 | 37.1 | 0.361 |
| StarGAN-v2 | 0.89 | 0.75 | 19.8 | 0.432 |
| FUNIT | 0.94 | 0.85 | 18.8 | 0.436 |
| Ours w/o style modeling | 0.80 | 0.79 | 55.9 | 0 |
| Ours w/o adversarial loss | **0.33** | **0.38** | 29.1 | 0.448 |
| Ours | **0.33** | 0.42 | **16.5** | **0.511** |
| Optimal | 0.33 | 0.33 | 12.9 | - |

## 4.1 BASELINES

**Multi-modal Domain Translation** We compare to StarGAN-v2 and FUNIT which are the state-of-the-art domain translation methods and the established MUNIT, DRIT and MSGAN frameworks (which must be trained multiple times for every possible pair of domains). As standard LORD does not support multi-modal settings, we use its ImageNet K-MEANS augmented setting.

**Class-supervised Disentanglement** We compare to LORD and FUNIT as they can handle fine-grained class labels (i.e. $10k$ face identities), and generalize to unseen classes.

**Single Attribute Manipulation** We further assess the performance of our disentanglement approach in the task of translating Males to Females against FaderNetworks (Lample et al., 2017) and mGAN-prior (Gu et al., 2020) which operates in the latent space of StyleGAN.

## 4.2 DATASETS

**AFHQ (Choi et al., 2020)** $15,000$ high quality images categorized into three domains: cat, dog and wildlife. We follow the protocol used in StarGAN-v2 and use the images at $256 \times 256$ resolution, holding out 500 images from each domain for testing.

**CelebA (Liu et al., 2015)** 202,599 images of 10,177 celebrities. We designate the person identity as class. We crop the images to $128 \times 128$ and use 9,177 classes for training and 1,000 for testing.

**CelebA-HQ (Karras et al., 2018)** 30,000 high quality images from CelebA. We set the gender as class. We resize the images to $256 \times 256$ and leave 1,000 images from each class for testing.

## 4.3 EVALUATION PROTOCOL

We assess the disentanglement at two levels: the learned representations and the generated images. At the representation level, we follow the protocol in LORD (Gabbay & Hoshen, 2020) and train a two-layer multi-layer perceptron to classify class labels from the learned content codes (lower accuracy indicates better disentanglement). In CelebA, where annotations of part of the common attributes are available (for evaluation only) such as 68-facial landmark locations, we train a linear regression model to locate the landmarks given the learned class (identity) codes (higher error indicates better disentanglement). At the image level, we follow StarGAN-v2 and translate all images in the test set to each of the other domains multiple times, borrowing style codes from random reference images in the target domain. We then train a classifier to classify the original class of the content image from the generated image. A lower accuracy indicates better disentanglement as the source class does not leak into the translated image. We also compute FID (Heusel et al., 2017) in

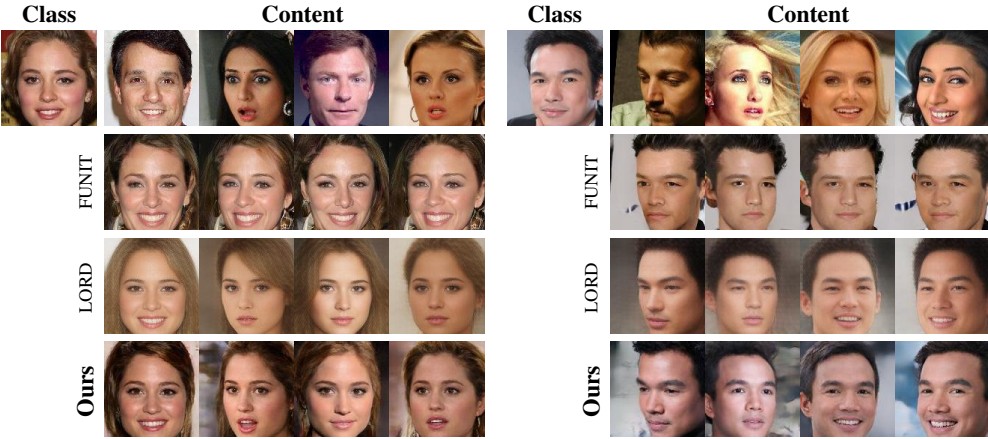

Figure 4: CelebA: Class (i.e. facial identity) and style (e.g. illumination) are guided by the reference image, while the content (e.g. head pose and expression) of the source image should remain intact. FUNIT preserves head pose but fails to model facial expression or transfer the exact facial identity. LORD captures the facial identity but generates low fidelity images. Our method preserves head pose and facial expression well while transferring the identity and generating appealing images.

Table 2: Class-supervised disentanglement of 10K classes on CelebA. Representation results: classification accuracy of class labels from content codes ($y \leftarrow c$) and the error of landmarks regression from class codes ($R(e_y) \rightarrow c$). Identity = FaceNet cosine similarity, Expression = RMSE on facial landmarks, Head Pose = yaw, pitch, roll angle errors, Rec = reconstruction error (LPIPS). * Indicates loss of head pose and expression details.

| | Representation | | Image | | | |
|---|---|---|---|---|---|---|
| | $y \leftarrow c \downarrow$ | $R(e_y) \rightarrow c \uparrow$ | Identity $\uparrow$ | Expression $\downarrow$ | Head Pose $\downarrow$ | Rec $\downarrow$ |
| FUNIT | 0.16 | 2.6 | 0.24 | 3.82 | 5.2, 6.9, 2.0 | 0.334 |
| LORD | **0.0001*** | **3.6** | 0.48 | 3.17 | 4.2, 4.6, 1.8 | 0.325 |
| Ours | **0.001** | **3.6** | **0.63** | **2.71** | **2.8, 3.3, 1.4** | **0.310** |
| Optimal | 0.001 | - | 1 | 0 | 0, 0, 0 | 0 |

a conditional manner to measure the discrepancy between the distribution of images in each target domain and the corresponding translations generated by the models. A lower FID score indicates that the translations are more reliable and better fit to the target domain. FID between real train and test images of the same class forms the optimal score for this metric. In order to assess the diversity of translation results, we measure the perceptual pairwise distances using LPIPS (Zhang et al., 2018) between all translations of the same input image. Higher average distances indicate greater diversity in image translation. In cases where external annotation methods are available (for evaluation only), such as face recognition (Cao et al., 2018) and head pose (Ruiz et al., 2018) and landmark detection for CelebA, we further measure the similarity of the identity of the generated face and the reference, as well as expression (represented by landmarks) and head pose errors. While we do not have ground truth for translation in CelebA as well, we present a reconstruction error (LPIPS similarity) by inferring the class (identity) representation from one image and reconstructing a second of the same person by extracting the content representation from an image of a different identity which has the closest pose (nearest neighbour in the 68 facial-landmarks space). For the single attribute manipulation experiment on CelebA-HQ, we measure the accuracy of fooling a target classifier.

## 4.4 RESULTS

We present a quantitative evaluation of multi-modal domain translation on AFHQ in Tab. 1. Our method outperforms all baselines achieving near perfect disentanglement at the representation level

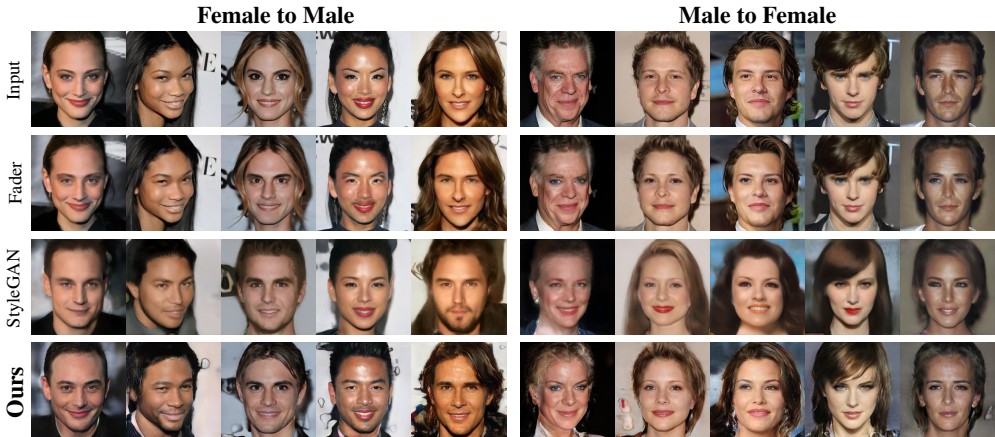

Figure 5: CelebA-HQ Male/Female: Our method makes more significant changes to the face than Fader Networks while preserving the similarity to the original identity better than mGANprior.

Table 3: Male-to-Female translation results on CelebA-HQ.

|  | Target Classification ↑ | | FID ↓ | |
| --- | --- | --- | --- | --- |
|  | F2M | M2F | F2M | M2F |
| Fader Networks | 0.82 | 0.80 | 119.7 | 81.7 |
| mGANprior on StyleGAN | 0.59 | 0.76 | 78.2 | 45.3 |
| Ours | **0.98** | **0.97** | **54.0** | **42.9** |
| Optimal (Real Images) | 0.99 | 0.99 | 15.6 | 14.0 |

as the classifier could barely guess class labels from content codes by a random chance. Similar results are obtained by trying to classify the original class of the translated image. We hypothesize that the minor degradation in this metric is due to using a deeper classifier at the image level (VGG-based). It can be seen that the baselines leak a little bit less information at the output image compared to the representation level, which might indicate that their generator ignores some parts of the representations. Moreover, we achieve both better visual quality (FID) and higher translation diversity (LPIPS). As can be seen in the qualitative examples in Fig. 3, StarGAN-v2 and FUNIT leak a significant amount of details from the input image which results in unreliable translations. Our method generates much more disentangled translations and captures the exact style of the reference image. In order to assess the performance in cases where fine-grained class labels are available and the ability to generalize to unseen classes at inference time is essential, we provide comparison on CelebA with 10K different identities in Tab. 2. It can be seen that our method achieves better expression and pose transfer while preserving the face identity more faithfully than the baselines. To validate that content is distributed evenly across identities, we use landmark annotations together with pose-related attributes from CelebA (Open Mouth and Smiling) and train a classifier to infer the identity. The accuracy of this classifier (0.001) forms the optimal result for the representation metric. Examples from this experiment are presented in Fig. 4. We also assess the performance of our method in the single attribute manipulation task of translating males to females on CelebA-HQ in Tab. 3 and Fig. 5. FaderNetworks makes minor changes to the given face which do not result in reliable translations. mGANprior modifies the input face significantly which reduces the similarity to the original identity, as many subtle facial attributes are lost when inverting real images within the StyleGAN latent space. Moreover, some of the images were not well reconstructed even though StyleGAN is trained on CelebA-HQ. Our model achieves near perfect classification score and generates visually pleasing results. More qualitative results are provided in Appendix A.2.

## 4.5 ABLATION STUDY

In order to isolate the contribution of our main principles, we provide an assessment of several variants of our method.

**Style modeling for multi-modal translation**    We train our model without style modeling (treating content and style as a single representation, both are complementary to the class). As can be seen in Tab. 1, this strategy violates the disentanglement criteria as class-specific attributes are now modeled by the content. As a consequence, the animals are translated in an unreliable and entangled manner between classes, as can be seen in Fig.11 in Appendix A.4.

**Adversarial loss for perceptual quality**    We train our method without the adversarial loss in the synthesis stage. The results in Tab. 1 suggest that the disentanglement criteria are achieved by our non-adversarial framework, while the additional adversarial loss contributes for increasing the output fidelity (FID). Qualitative evidence for this claim is presented in Fig.11 in Appendix A.4.

**Limitations**    There are two main limitations of the proposed method; i) The disentanglement between content and style (both are complementary to the class) is achieved in an unsupervised manner and therefore relies on an inductive bias. In our framework it is driven by the design of a set of random transformations to which the style needs to be invariant. While we show that common settings in image translation can be solved effectively with simple operators, other settings may require more sophisticated transformations. ii) Our model is not well optimized for cases in which the object is not the major part of the image e.g. where the background contains other objects or large variation. We hypothesize that this can be better solved by introducing an unsupervised segmentation objective into the entire framework. We leave this direction to future research.

## 5 CONCLUSION

We present a principled approach for learning disentangled representations in a non-adversarial framework, leveraging latent optimization and well-motivated content and style bottlenecks. We show how adversarial optimization can be decoupled from representation disentanglement and be applied only to increase the fidelity of the generated images. Our proposed model learns disentangled representations and achieves state-of-the-art performance compared to both adversarial and non-adversarial disentanglement methods in different settings of image translation.

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

## A   APPENDIX

### A.1   IMPLEMENTATION DETAILS

**Architectures**   We borrow the generator and discriminator architectures from StyleGAN2 with two modifications; (i) The latent code is adapted to our formulation and forms a concatenation of three separate latent codes: content, class and style. (ii) We do not apply any noise injection during training. A brief summary of the architectures is presented in Tab. 4 and 5 for completeness. The architectures of the content, class and style encoders is influenced by StarGAN-v2 (Choi et al., 2020) and presented in Tab. 6. Note that we do not use any domain-specific layers and our model is able to generalize to unseen domains at inference time. For cases where the style is minimal and should mainly model low-level image features (e.g. the information that is not described by identity and pose in CelebA) we use a shallow style encoder with a small receptive field, as can be seen in Tab. 7.

**Optimization**   In the first stage, we optimize over a single content embedding per image (content dim = 64), a single class embedding per class (class dim = 512) and the parameters of the style encoder (style dim = 64) and the generator. We set the learning rate of the latent codes to 0.01, the learning rate of the generator to 0.001 and the learning rate of the style encoder to 0.0001. The penalty of the content bottleneck is set to $\lambda_{cb} = 0.001$. We train the disentanglement stage for 200 epochs. For each mini-batch, we update the parameters of the models and the latent codes with a single gradient step each. In the synthesis stage, we add two encoders to infer the content and class codes learned in the first stage directly from an image. We also optimize a discriminator in an end-to-end manner to increase the perceptual fidelity of the images. This stage is trained for 100 epochs and the learning rate for all the parameters in this stage is set to 0.0001, except for the style encoder whose weights are kept fixed.

**Transformations**   As stated in Sec. 3, the design of the transformation applied on the input to the style encoder is setting dependent. For AFHQ, in order to encourage the style to be invariant to the head pose of the animal, we randomly flip the image horizontally and crop 0.6 to 1 portion of the image. In CelebA, as we are given fine grained class labels (person identity), the desired style should ideally model only low level features and therefore we opt for a shallow style encoder with a small receptive field (5x5) except for the same augmentations as of AFHQ. The low receptive field avoids the style from encoding common-attributes as facial expression (which might retain even after the transformation).

**Baseline models**   For the evaluation of competing methods, we use the following official publicly available pretrained models: StarGAN-v2 (for AFHQ) and StyleGAN (for mGANprior on CelebA-HQ). We train the rest of the baselines using the official repositories of their authors and make an effort to select the best configurations available for the target resolution (for example, FUNIT trained by us for AFHQ achieves similar results to the public StarGAN-v2 which was known as the SOTA on this benchmark).

Table 4: Generator architecture based on StyleGAN2. StyleConv and ModulatedConv use the injected latent code composed of the content, class and style representations.

| Layer | Kernel Size | Activation | Resample | Output Shape |
|---|---|---|---|---|
| Constant Input | - | - | - | $4 \times 4 \times 512$ |
| StyledConv | $3 \times 3$ | FusedLeakyReLU | - | $4 \times 4 \times 512$ |
| StyledConv | $3 \times 3$ | FusedLeakyReLU | UpFirDn2d | $8 \times 8 \times 512$ |
| StyledConv | $3 \times 3$ | FusedLeakyReLU | - | $8 \times 8 \times 512$ |
| StyledConv | $3 \times 3$ | FusedLeakyReLU | UpFirDn2d | $16 \times 16 \times 512$ |
| StyledConv | $3 \times 3$ | FusedLeakyReLU | - | $16 \times 16 \times 512$ |
| StyledConv | $3 \times 3$ | FusedLeakyReLU | UpFirDn2d | $32 \times 32 \times 512$ |
| StyledConv | $3 \times 3$ | FusedLeakyReLU | - | $32 \times 32 \times 512$ |
| StyledConv | $3 \times 3$ | FusedLeakyReLU | UpFirDn2d | $64 \times 64 \times 512$ |
| StyledConv | $3 \times 3$ | FusedLeakyReLU | - | $64 \times 64 \times 512$ |
| StyledConv | $3 \times 3$ | FusedLeakyReLU | UpFirDn2d | $128 \times 128 \times 256$ |
| StyledConv | $3 \times 3$ | FusedLeakyReLU | - | $128 \times 128 \times 256$ |
| StyledConv | $3 \times 3$ | FusedLeakyReLU | UpFirDn2d | $256 \times 256 \times 128$ |
| StyledConv | $3 \times 3$ | FusedLeakyReLU | - | $256 \times 256 \times 128$ |
| ModulatedConv | $1 \times 1$ | - | - | $256 \times 256 \times 3$ |

Table 5: Discriminator architecture based on StyleGAN2.

| Layer | Kernel Size | Activation | Resample | Output Shape |
|---|---|---|---|---|
| Input | - | - | - | $256 \times 256 \times 3$ |
| Conv | $3 \times 3$ | FusedLeakyReLU | - | $256 \times 256 \times 128$ |
| ResBlock | $3 \times 3$ | FusedLeakyReLU | UpFirDn2d | $128 \times 128 \times 256$ |
| ResBlock | $3 \times 3$ | FusedLeakyReLU | UpFirDn2d | $64 \times 64 \times 512$ |
| ResBlock | $3 \times 3$ | FusedLeakyReLU | UpFirDn2d | $32 \times 32 \times 512$ |
| ResBlock | $3 \times 3$ | FusedLeakyReLU | UpFirDn2d | $16 \times 16 \times 512$ |
| ResBlock | $3 \times 3$ | FusedLeakyReLU | UpFirDn2d | $8 \times 8 \times 512$ |
| ResBlock | $3 \times 3$ | FusedLeakyReLU | UpFirDn2d | $4 \times 4 \times 512$ |
| Concat stddev | $3 \times 3$ | FusedLeakyReLU | UpFirDn2d | $4 \times 4 \times 513$ |
| Conv | $3 \times 3$ | FusedLeakyReLU | - | $4 \times 4 \times 512$ |
| Reshape | - | - | - | 8192 |
| FC | - | FusedLeakyReLU | - | 512 |
| FC | - | - | - | 1 |

**Training resources** Training takes approximately 3 days for 256x256 resolution on a single NVIDIA RTX-2080 TI.

## A.2 Additional results

We provide additional qualitative results on AFHQ in Fig. 6,7, on CelebA in Fig. 8, 9 and on CelebA-HQ in Fig. 10.

## A.3 Latent optimization

In this work, we opt for learning the content representation using latent optimization, similarly to LORD (Gabbay & Hoshen, 2020). Autoencoders assume a parametric model, usually referred to as the encoder, to compute a vector $c_i$ from image $x_i$. on the other hand, we jointly optimize the inputs and the model parameters. Since the vector $c_i$ is learned directly and is unconstrained by a parametric encoder function, our model can recover all the solutions that could be found by an autoencoder, and reach some others. In order to justify this design choice, we validate the observation presented in LORD and train our disentanglement stage in an amortized fashion using a content encoder. As can be seen in Fig. 14, amortized training fails to reduce the correlation between the content and class representations. We have experimented with several content decay factors ($\lambda_{cb}$ in Eq. 5). Although

Table 6: Encoder architecture based on StarGAN-v2. Note that we do not use any domain-specific layers. $D$ is the dimension of the content, class or style code respectively.

| Layer | Kernel Size | Activation | Resample | Output Shape |
|---|---|---|---|---|
| Input | - | - | - | $256 \times 256 \times 3$ |
| Conv | $3 \times 3$ | - | - | $256 \times 256 \times 64$ |
| ResBlock | $3 \times 3$ | LeakyReLU ($\alpha = 0.2$) | Avg Pool | $128 \times 128 \times 128$ |
| ResBlock | $3 \times 3$ | LeakyReLU ($\alpha = 0.2$) | Avg Pool | $64 \times 64 \times 256$ |
| ResBlock | $3 \times 3$ | LeakyReLU ($\alpha = 0.2$) | Avg Pool | $32 \times 32 \times 256$ |
| ResBlock | $3 \times 3$ | LeakyReLU ($\alpha = 0.2$) | Avg Pool | $16 \times 16 \times 256$ |
| ResBlock | $3 \times 3$ | LeakyReLU ($\alpha = 0.2$) | Avg Pool | $8 \times 8 \times 256$ |
| ResBlock | $3 \times 3$ | LeakyReLU ($\alpha = 0.2$) | Avg Pool | $4 \times 4 \times 256$ |
| Conv | $4 \times 4$ | LeakyReLU ($\alpha = 0.2$) | - | $1 \times 1 \times 256$ |
| Reshape | - | - | - | $256$ |
| FC | - | - | - | $D$ |

Table 7: Encoder shallow architecture with low receptive field for settings in which classes only exhibit a low level intra style variation (e.g. CelebA).

| Layer | Kernel Size | Activation | Resample | Output Shape |
|---|---|---|---|---|
| Input | - | - | - | $256 \times 256 \times 3$ |
| Conv | $3 \times 3$ | LeakyReLU ($\alpha = 0.2$) | - | $256 \times 256 \times 64$ |
| Conv | $3 \times 3$ | - | Global Avg Pool | $64$ |
| FC | - | - | - | $64$ |

the disentanglement improves as $\lambda_{cb}$ increases, the reconstruction gets worse and the model fails to converge with $\lambda_{cb} > 0.1$.

## A.4 VISUALIZATION OF ABLATION ANALYSIS

Examples from the ablation analysis are provided in Fig. 11. Visualization of the three factors modeled by our method is provided in Fig. 12 and Fig. 13.

## A.5 FAILURE CASES

As our method strongly relies on the style image to infer the class-specific attributes e.g. identity and facial shape, we have found that when the animal in the style image is present in an extreme pose such as wide open mouth, the style encoder can sometime confuse the extreme pose as a property of the breed and transfer it from the style image. Examples are provided in Fig. 15.

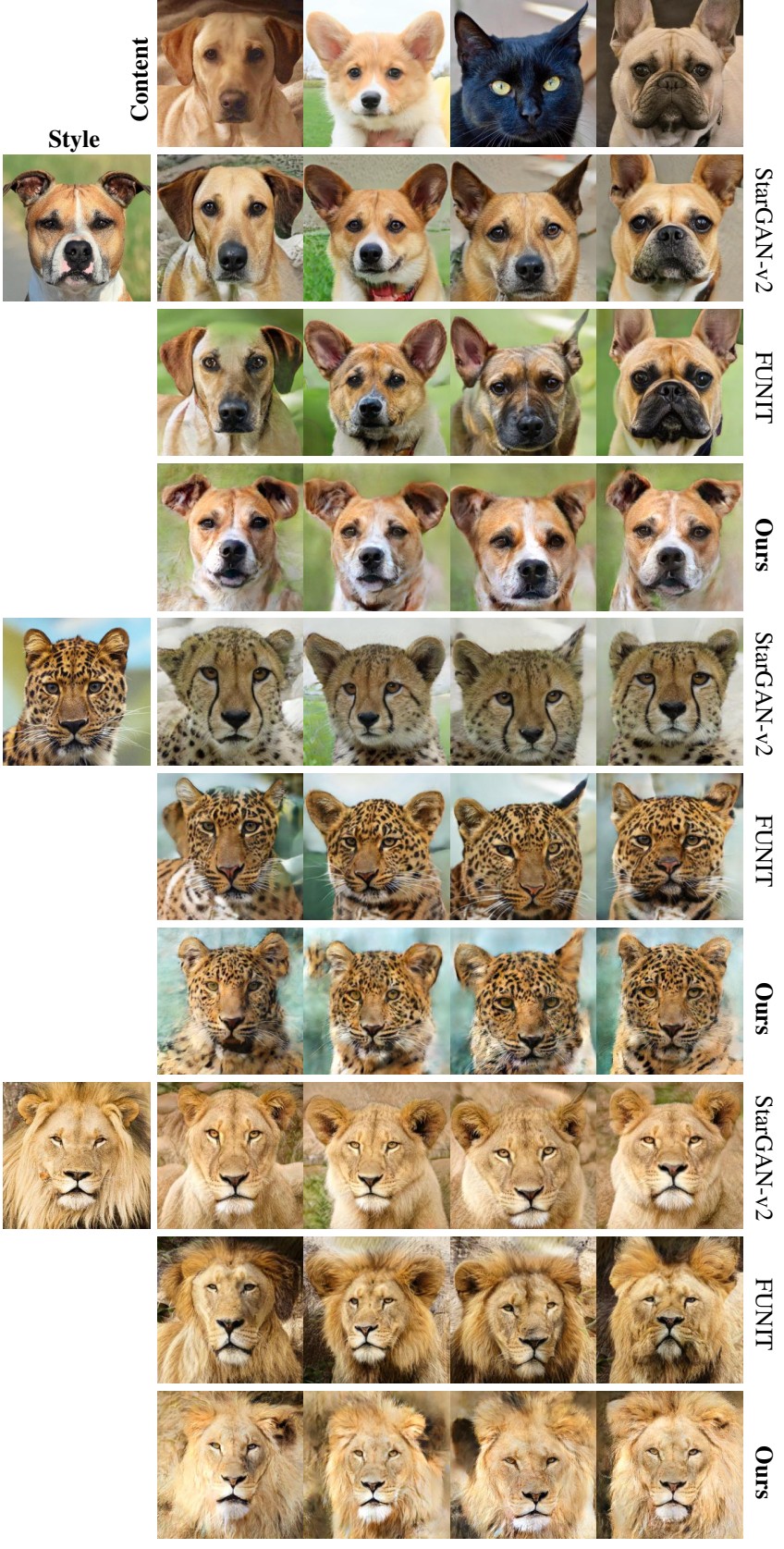

Figure 6: More qualitative results on AFHQ.

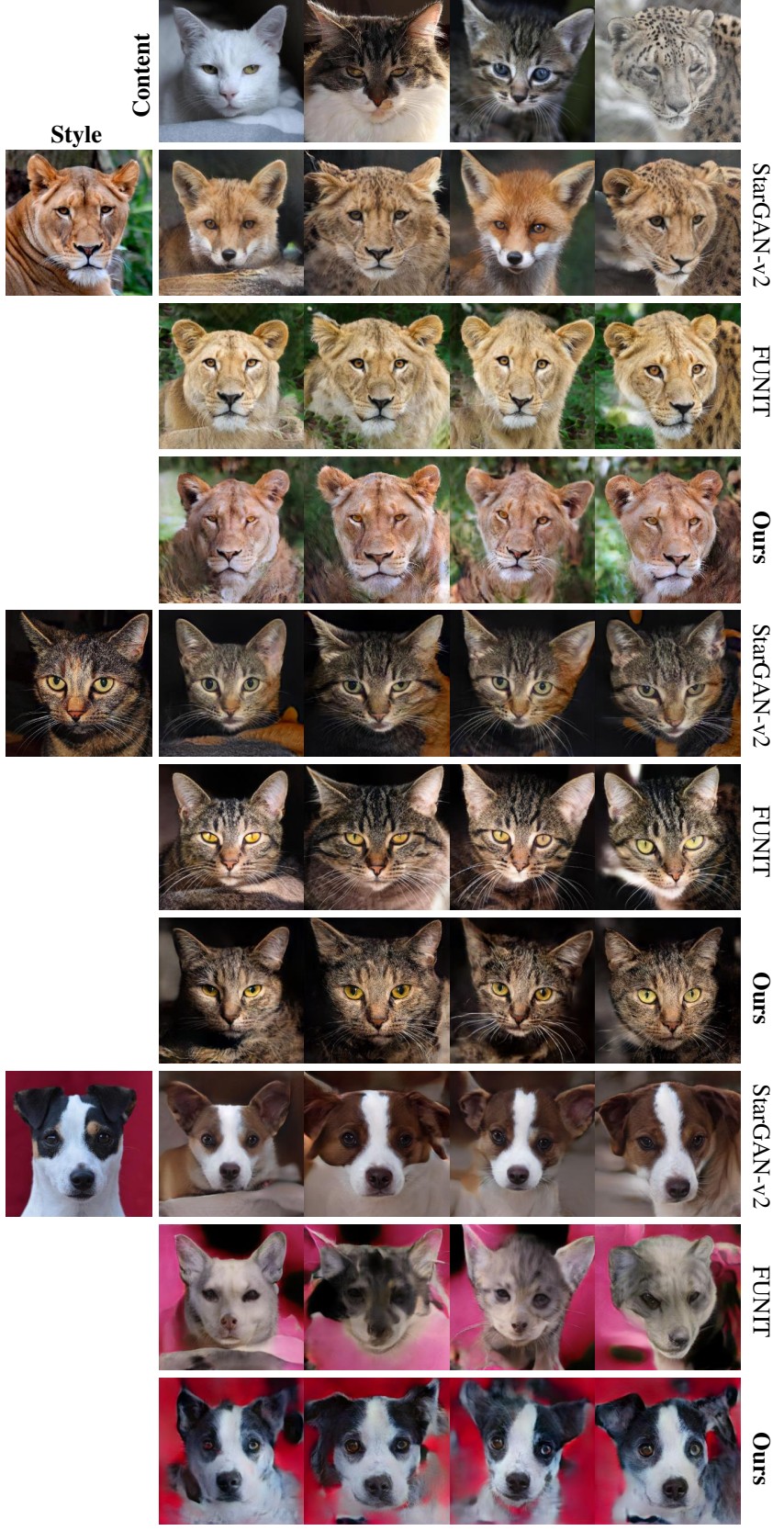

Figure 7: More qualitative results on AFHQ.

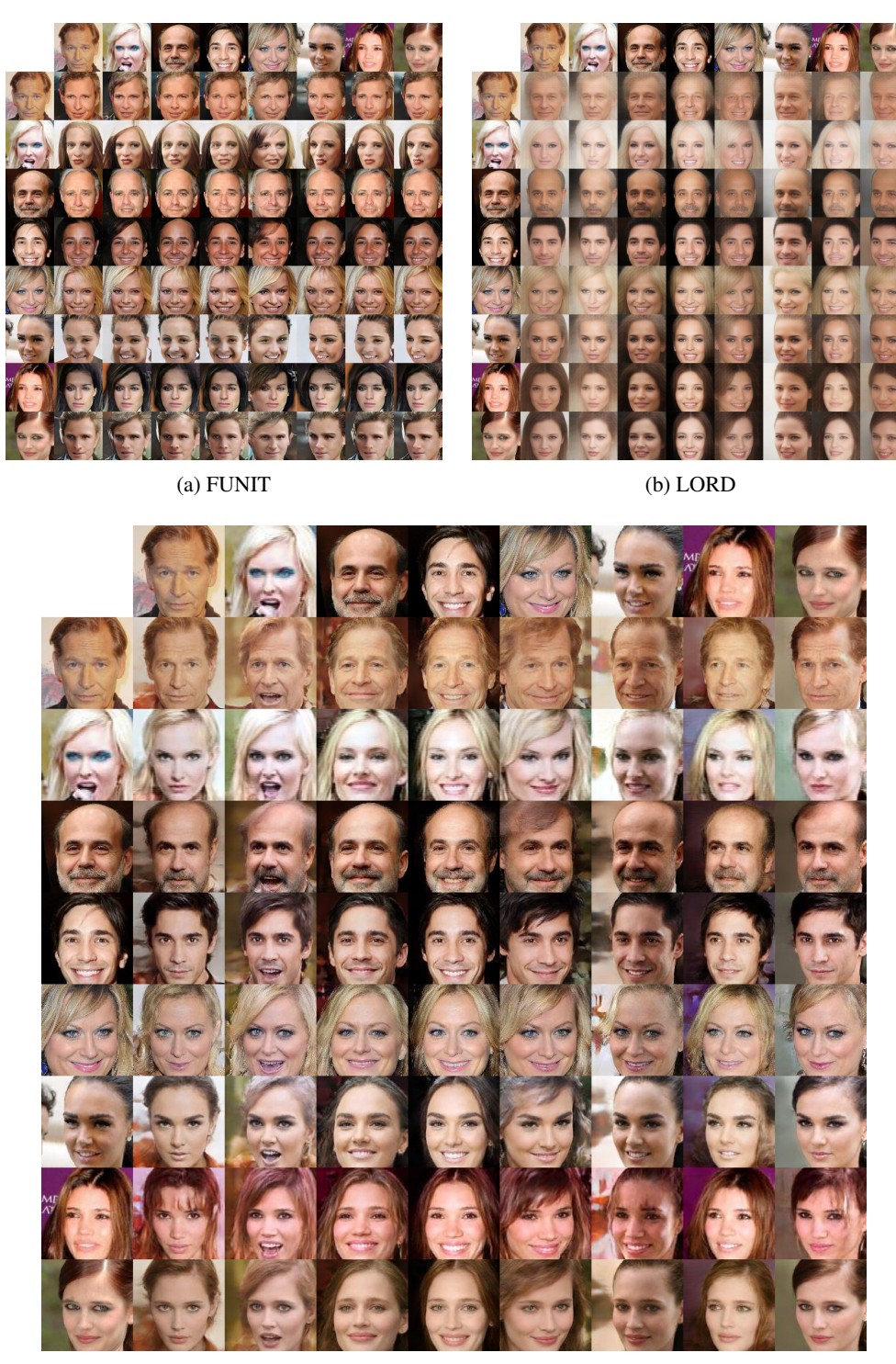

(a) FUNIT

(b) LORD

(c) Ours

Figure 8: More qualitative results on CelebA. Top row: Original content images. Left column: Original class and style images.

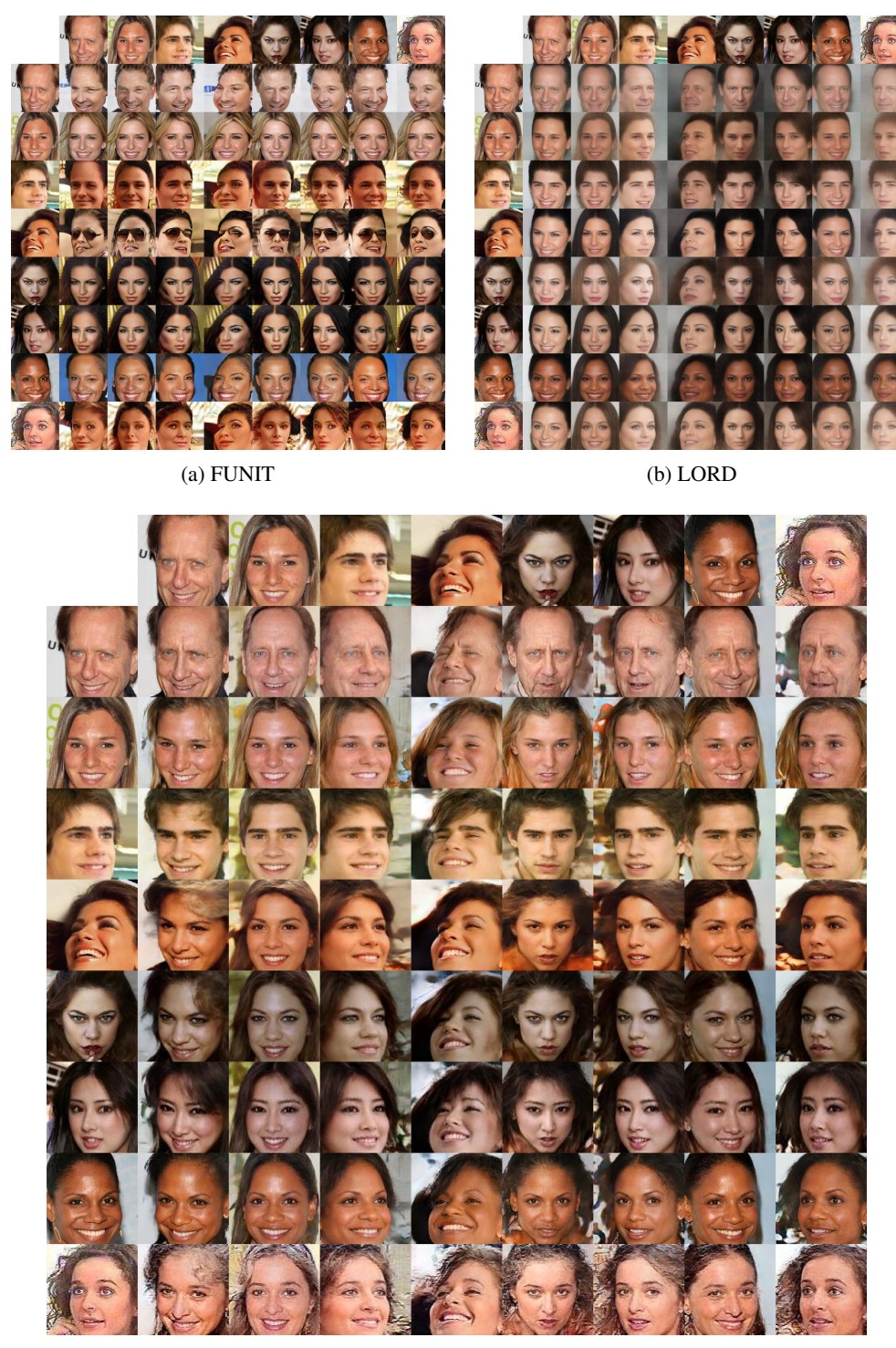

(a) FUNIT

(b) LORD

(c) Ours

Figure 9: More qualitative results on CelebA. Top row: Original content images. Left column: Original class and style images.

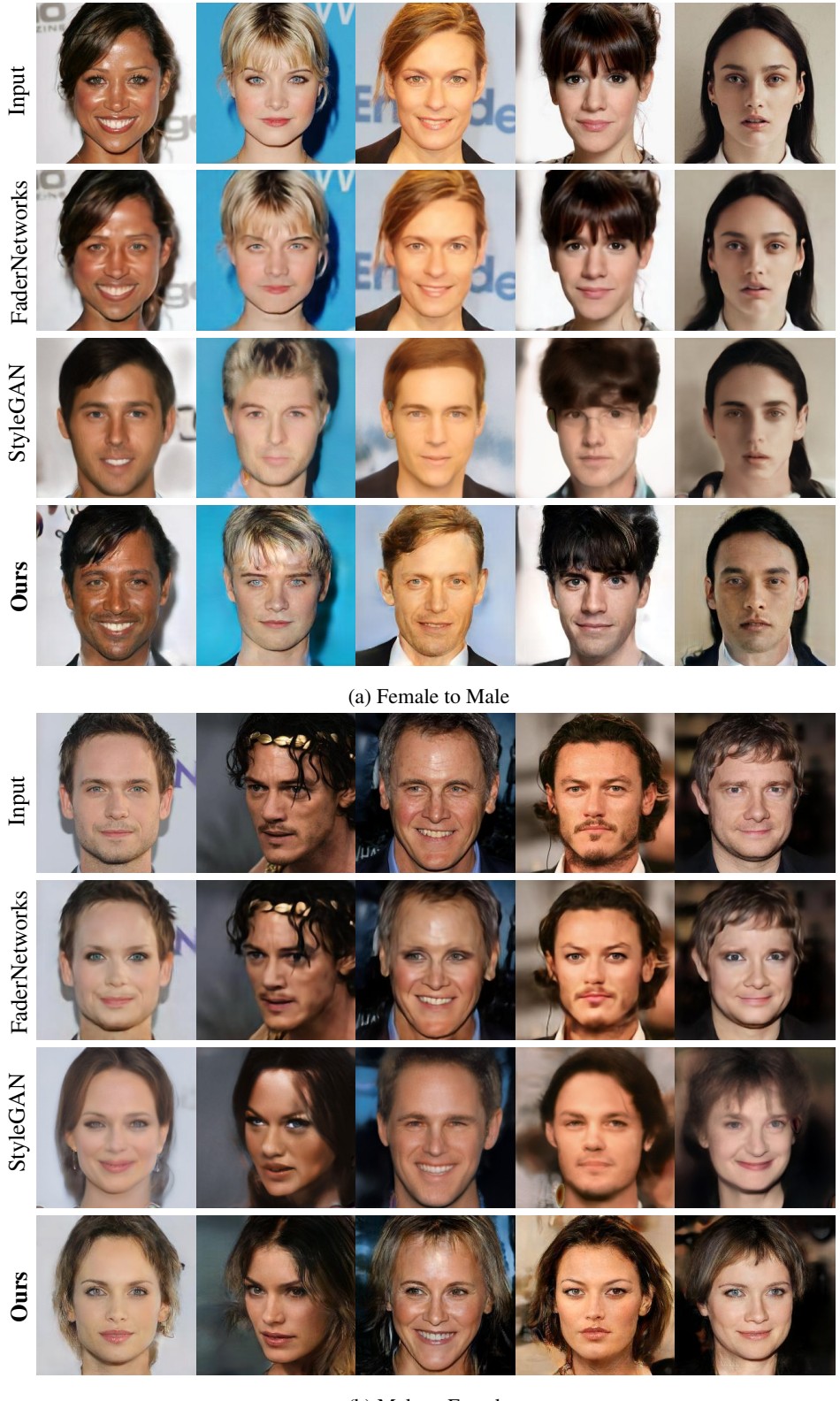

(a) Female to Male

(b) Male to Female

Figure 10: More qualitative results on CelebA-HQ.

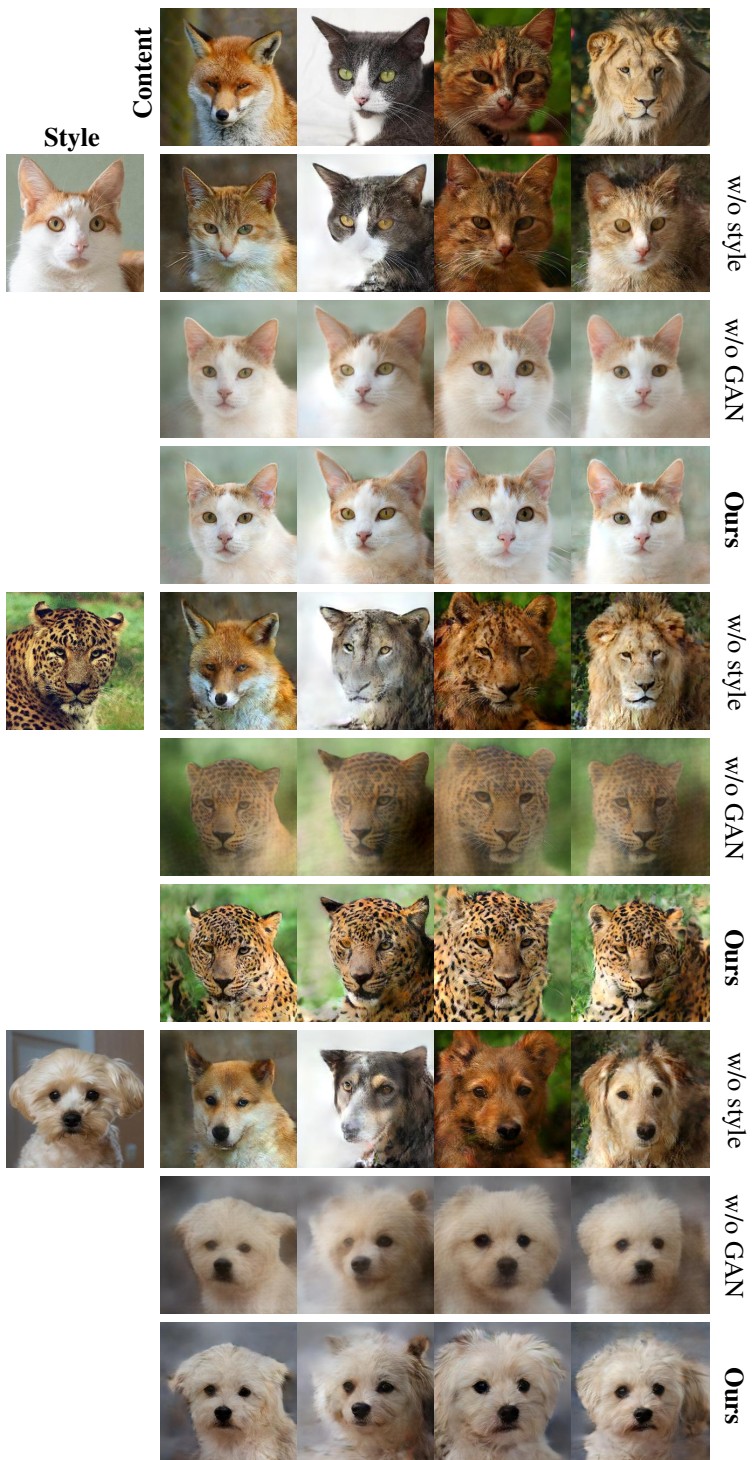

Figure 11: Qualitative examples from the ablation analysis; (i) Only a single translation is possible without modeling style, which often yields entangled results. (ii) Disentanglement is achieved without adversarial loss. (iii) Adversarial loss contributes to visual fidelity.

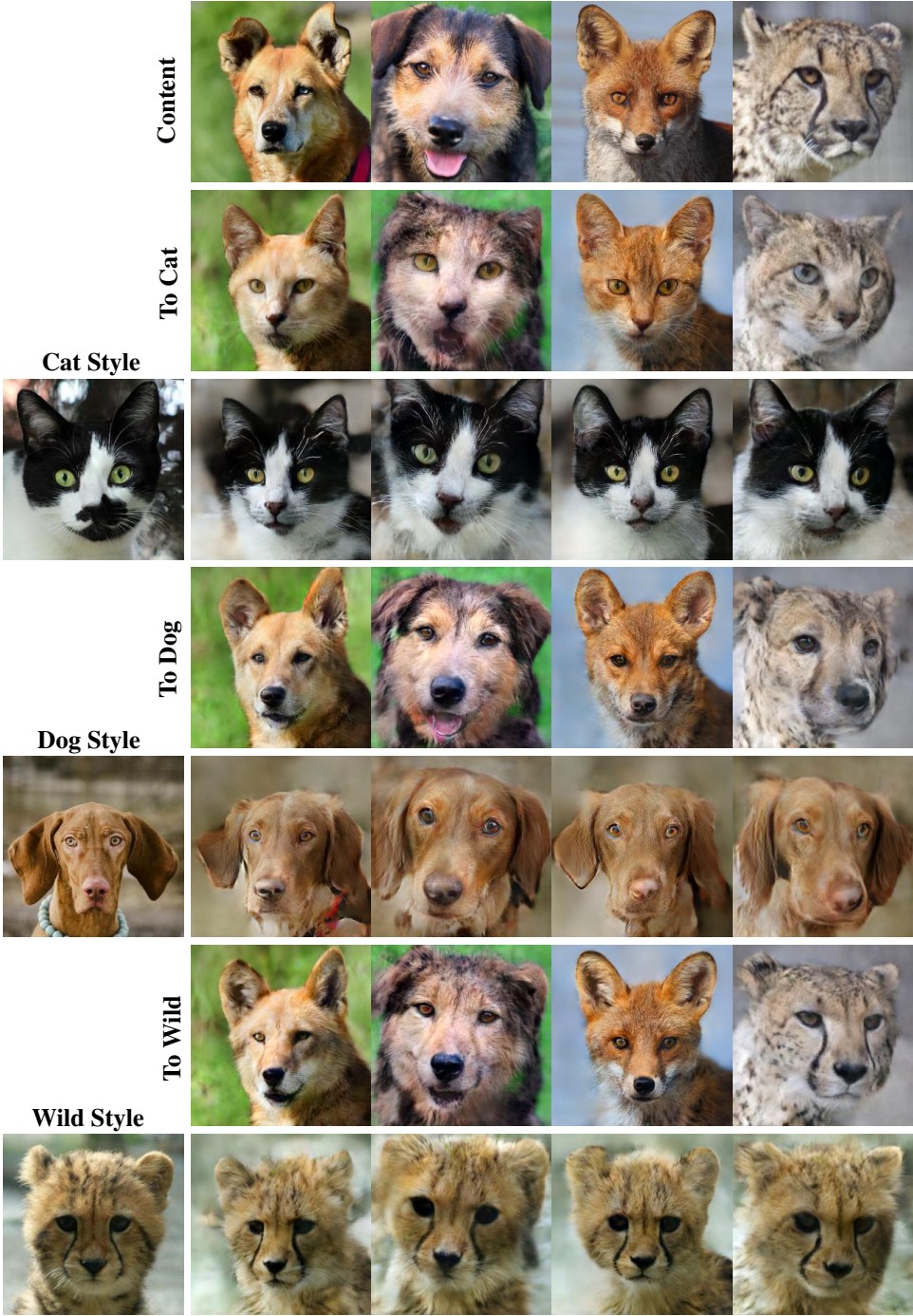

Figure 12: Visualization of the three factors modeled by our method. Changing the class (e.g. To cat, dog, wild) while leaving the style intact affects high level semantics of the presented animal. Borrowing the style from a reference image allows for specification of the exact target breed. Note that the class and style are correlated and therefore transferring style across classes can generate unreliable translations (e.g. first two images in "To Wild".)

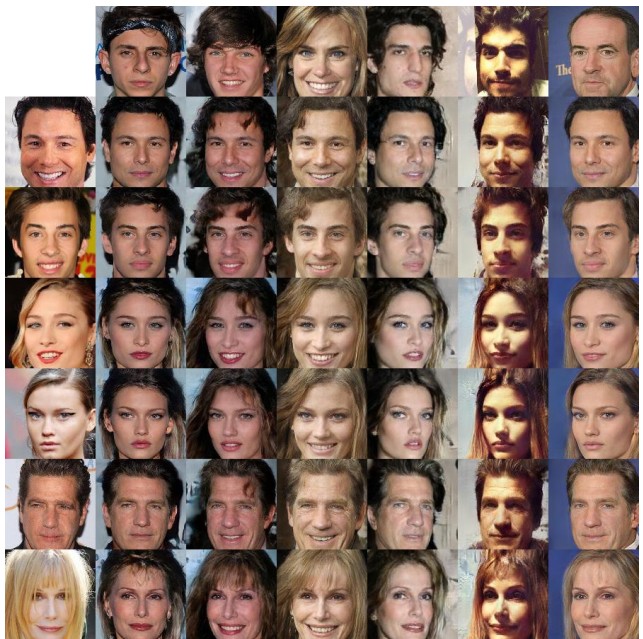

(a) Transferring class from images on the left column to images on the top row.

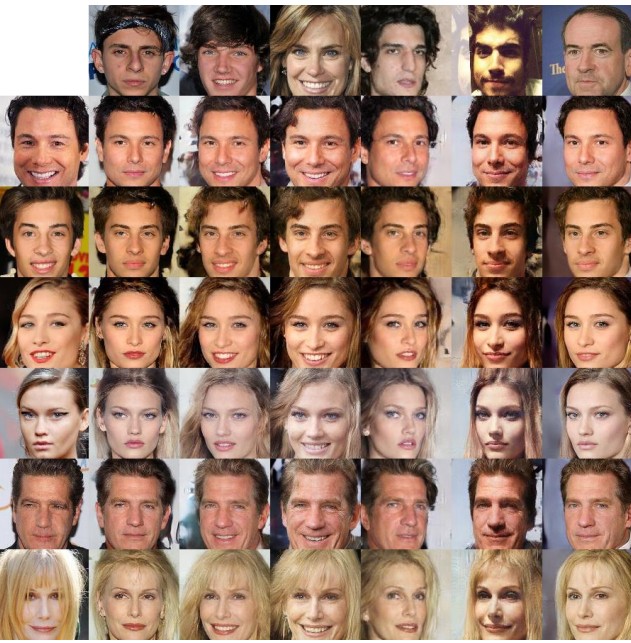

(b) Transferring class and style from images on the left column to images on the top row.

Figure 13: Visualization of the three factors modeled by our method. Top row and left column present original images. Translating the images in each column to the class of the image on the left column (a) affects the person identity. Borrowing the style from the image on the left column (b) transfers illumination and class-specific attributes as hair color. In both cases the head pose and the facial expression remain intact.

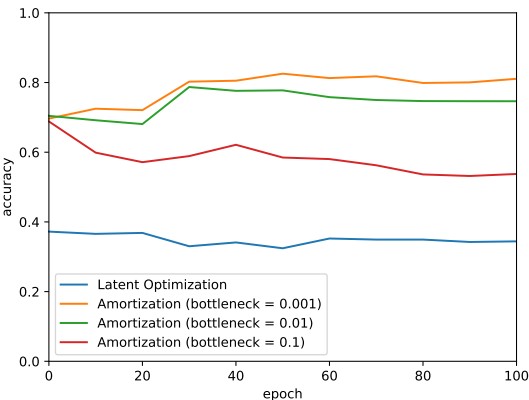

Figure 14: Evidence for the inductive bias conferred by latent optimization on AFHQ (a validation of the discovery presented in Gabbay & Hoshen (2020)). Latent optimization starts with random initialized content codes and preserves the disentanglement between content and class along the entire training. In contrast, a randomly initialized content encoder outputs entangled codes. In order to result in disentangled content codes, it is required to remove the class information from the content during the course of optimization. In this plot we can see that the optimization is in practice unsuccessful at doing so.

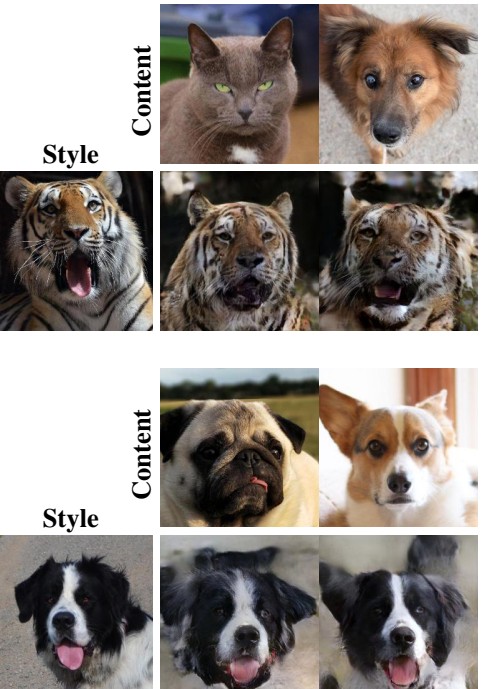

Figure 15: Failure cases where the mouth and tongue are transferred over from the style image.

