# OpenReview forum: "Learning Disentangled Representations for Image Translation"
_ICLR.cc/2021/Conference — Reject_

### Official Review · AnonReviewer3 · 2020-10-29
**Confusing about the difference of style and class.**

**Rating:** 4
**Confidence:** 3

**Review:**

This paper proposes OverLORD for learning disentangled representations for image translation. The method employs a latent optimization framework and followed a GAN training process. This work also designs a style encoding framework by introducing transformation before style encoder. The results show good fidelity performance.

Apparently, this work is inspired by LORD, another non-adversarial method of learning disentangled representation. The concepts of class and content are borrowed from the LORD algorithm. This paper introduces an additional disentanglement dimension called style, which seems to be learned unsupervised. The learning method for this style is similar to learning general attributes that are robust to transformations.

My major concern is also about this ''style''. From Figure.3 and Figure.4, what is the different between ''style'' and ''class''? It seems that ''style'' and ''class'' are somewhat coupled. Could the author please explain what is the ''style'', ''common attributes'', ''class'' and ''class specific attributes'' for both cases in Figure.3 and 4. In order to enable readers to better understand, authors can consider visual transformations and the learned styles.

The second stage employs adversarial training, which greatly improved the visual effect. But the introduction of adversarial training is not a huge novelty. In the comparison, I am curious how much better OverLORD without adversarial training is than LORD. Is the introduction of style codes what makes OverLORD more suitable for image translation? If it is, please verify it experimentally.

If the authors are able to address the above questions, then I am happy to raise my score.

---

> ### Author Response · Authors · 2020-11-14
> **Response to Reviewer #3**
>
> We thank the reviewer for the dedicated review and for recognizing the “good fidelity performance” of our method. The reviewer raised several valid points, which we believe can be easily addressed.
>
> **“What is the difference between ''style'' and ''class''? It seems that ''style'' and ''class'' are somewhat coupled. Could the author please explain what is the ''style'', ''common attributes'', ''class'' and ''class specific attributes'' for both cases in Figure.3 and 4”**: We clarified the role of each representation in the captions of the figures. In both figures, the class and styles are guided together by a reference image, as this is the most meaningful translation in most cases. To better understand the impact of each factor separately, we provide a visualization in Fig. 12 and Fig. 13 in Appendix A.4.
>
> In general, the difference between class and style may seem subtle but it is crucial in various disentanglement settings. Assume the image is specified by three factors: animal type (class) e.g. dog, cat or wild, breed (style) e.g. “poodle” and pose (content). As we are given a single label (representing the class) per image, we learn all the attributes supervised by the class label as a class representation. In AFHQ, the pose is independent of the class and can therefore be transferred. On the other hand, the breed of the animal is correlated to its class and therefore cannot always be transferred across classes. We find that in some cases changing the class while remaining the style intact does generate attractive results (see Fig. 12 and Fig. 13 in Appendix A.4).
>
> **“I am curious how much better OverLORD without adversarial training is than LORD. Is the introduction of style codes what makes OverLORD more suitable for image translation? If it is, please verify it experimentally”**: As LORD only supports class-content disentanglement, it does not allow the specification of the exact style of the target class (e.g. when translating a cat into a dog, there is no specification of the target dog breed we translate to). This translation formulation leads to the leakage of the class-specific attributes into the content representation. Table 1 shows evidence for this phenomenon in both LORD and an ablation of our method trained without style modeling. We further perform an exploratory experiment and reduce the amount of intra-class variation, when assessing the performance of LORD (denoted as LORD w/ style clustering in Table 1). We first cluster the images in each of the classes (cat, dog, wild) to 512 pseudo-classes using K-MEANS on VGG features trained on ImageNet. This preliminary step improves LORD performance to an extent. Finally, in order to emphasize the superiority of OverLORD compared to LORD (and the extended LORD), we train our method without the adversarial loss in the synthesis stage. The results in Table 1 suggest that a high level of disentanglement is achieved by our non-adversarial framework, outperforming LORD due to the style modeling approach. The additional adversarial loss further contributes to increasing the output fidelity (FID). Qualitative evidence for this claim is presented in Fig. 11 in Appendix A.4.
>
>
> We believe that all of the reviewer’s questions were addressed. The reviewer stated that addressing the questions would form a basis for raising the score.

---

### Official Review · AnonReviewer4 · 2020-10-30
**An approach to learn disentangled representation for image translation, with good qualitative results**

**Rating:** 6
**Confidence:** 5

**Review:**

Summary:

This paper proposes a novel approach named OverLORD to learn disentangled representations for image class and attributes. To tackle the problem of previous methods that the learned content and class are often entangled, the authors propose to disentangle image representations to class and attributes, and further disentangle attributes to common attributes among all classes (content), and class-specific attributes (style). It uses the idea from LORD to disentangle the image representations, and extends LORD to not only common attributes (content), but also class-specific attributes (style). In this way, it is able to transfer the common attributes while preserving the class and class-specific attributes. Experiments are conducted on animal faces and human faces datasets, and the proposed approach is able to preserve the class-specific attributes (e.g., shape or identity of the face) better than previous image-to-image translation methods that only disentangle style and content.

Pros:

1. By separating common attributes and class-specific attributes, the model achieves better control for image-to-image translation. For example, when conducting image-to-image translation for animal faces, previous approaches tend to encode both pose and shape/structure information in the content code, so that then transferring the pose of an animal face, the shape may also change. The disentanglement space in this paper gives clearer definitions and better control for image translation.

2. The proposed method uses a transformed version of the input images with random common attributes to force that the style encoder learns only class-specific style information. This is a smart design that successfully disentangles the class-specific attributes.

3. The results are pretty good compared with previous approaches.

Cons:

1. There are no ablation studies to validate the effectiveness of the architecture design. For example, (1) the effect of different loss functions. (2) What's the shape of the content embedding? Is it a feature vector or feature maps? Why do you choose the styleGAN2 structure? What if using a generator similar to the generator in FUNIT that uses AdaIN to combine the class and style information with the content information?

2. The random transformations for transforming the images before the style encoder are manually defined, and the set of transformations is different for different datasets (Appendix A.1). This restricts the approach from generalizing to other various datasets with different types of class-specific attributes.

3. The number of classes are predefined and fixed. It cannot generalize to unseen classes like FUNIT.

Reasons for score:

This paper solves a common problem in previous work for disentangling attributes and image translation. The method is quite novel and inspiring, and the result seems good compared with previous approaches.

Questions during the rebuttal period:

Please address and clarify the Cons above.

Post-rebuttal:

After reading the author feedback and other reviewers' comments, I would change my rating to 6. I partially agree with reviewer #3 and reviewer #5 that this work is an incremental extension of LORD. In addition, like I mentioned in the "Cons", the styles are decided by the pre-defined transformations that are dataset-specific. For different datasets the transformations are defined differently based on the properties of that dataset. This might restrict the method from being extended to new datasets with styles that are not easy to find corresponding transformations. However, I think this paper still have some inspiring aspects, like the idea of disentangling class-specific attributes and common attributes among all classes. And the results of this paper seems good compared with previous approaches. So I would give the rating of 6.

---

> ### Author Response · Authors · 2020-11-14
> **Response to Reviewer #4**
>
> We thank the reviewer for the dedicated and positive review and for finding our method “quite novel and inspiring”. The reviewer raised several valid points, which we believe can be easily addressed.
>
> **“No ablation studies to validate the effectiveness of the architecture design”**: We added two ablations of the two core components of our method with respect to LORD: We first train our model without style modeling (treating content and style as a single representation, both are complementary to the class). As can be seen in Table 1, this strategy violates the disentanglement as class-specific attributes are now modeled by the content. As a consequence, only a single translation is possible without modeling style, which often yields entangled results, as can be seen in Appendix A.4 Fig. 11 (w/o style). In order to assess the contribution of the additional adversarial loss, we train our method without the adversarial loss in the synthesis stage. The results in Table 1 suggest that a high level of disentanglement is achieved by our non-adversarial framework, while the additional adversarial loss contributes to increasing the output fidelity (FID). Qualitative evidence for this claim is presented in Appendix A.4 Fig. 11 (w/o GAN).
>
>
> **“What's the shape of the content embedding? Is it a feature vector or feature maps?“**: In order to avoid relying on inductive bias of a spatial form, we treat all the different representations as 1-D embeddings (See implementation details in Appendix A.1 for more details).
>
>
> **“Why do you choose the styleGAN2 structure? What if using a generator similar to the generator in FUNIT that uses AdaIN to combine the class and style information with the content information?”**: As we want to avoid any architectural biases which act differently on the different representations of class, content and style, we concatenate the three representations into a single 1-D embedding and feed it as a latent code to the generator. As state-of-the-art image translation architectures (e.g. StarGAN-v2, FUNIT) consist of biased modules e.g. spatial content representation (2d-feature map) with style injection using “AdaIN” layers (which inevitably constrains the model towards local changes) we opt for an unbiased StyleGAN-based architecture.
>
> **“This restricts the approach from generalizing to other various datasets with different types of class-specific attributes”**: The disentanglement of style and content is achieved in an unsupervised manner (as we only have supervision of the class). As an inductive bias of some form is essential, we opt for the design of a set of transformations applied to the style image.  While the exact set of transformations is indeed a setting dependent choice, we have found and shown in the paper that simple operators are suitable for the common configurations in image translation in which the content should ideally encode the pose of an object. Other settings may require more sophisticated transformations.
>
> **“The number of classes are predefined and fixed. It cannot generalize to unseen classes like FUNIT”**: We would like to bring to the reviewer’s attention that our method is able to generalize to unseen classes like FUNIT. For example, in the experiment on CelebA, we evaluate the performance of our model (and the baselines) on 1000 novel face identities that have been held out for testing. The training of the class encoder during the synthesis stage is designed exactly for this purpose.

---

### Official Review · AnonReviewer5 · 2020-11-07
**An extension of LORD (ICLR - 2020), which has great qualitative and quantitative results on style transferred image synthesis.**

**Rating:** 6
**Confidence:** 4

**Review:**

The paper presents a principled approach to style transfer by disentangling class-specific attributes from common (eq. class-independent) attributes. In order to do so, the paper leverages the formulation of a recently proposed disentangling approach called "LORD". The proposed approach is called OverLORD, and includes two main augmentations to LORD. The first is the introduction of a style encoder to learn a latent code for class-specific attributes, and the second is to introduce an adversarial learning in the second stage for high-quality style-transferred generation of images. The results are shown on three datasets: AFHQ (dog, cat, wildlife), CelebA (human faces), and CelebA-HQ (hi-res human faces), and are compelling in both qualitative and quantitative comparisons.

Strengths:
+ The weakness in LORD for style transfer applications has been identified and addressed (to an extent).
+ The evaluations and comparisons presented are thorough and the results are very compelling, both qualitative and quantitative. Many different metrics have been explored for evaluating both disentanglement as well as style-transferred image generation.

Weaknesses:
- The main contributions of this work seems to be handling the style and content disentangling, and the quality of image generation. While the results are impressive, the exposition does not justify why such an extension to LORD is a nontrivial contribution. Consequently, the paper seems like a "natural" (or straightforward) extension of LORD, giving an impression of limited novelty.
- LORD was ostensibly non-adversarial, but OverLORD is not. The necessity for an adversarial loss can be perhaps better justified by an ablation on with and without the adversarial loss to emphasize why a different kind of loss will not suffice.
- The disentangling between class and content essentially comes from the design of LORD's latent optimization. However, it is not clear what is the key factor for disentangling the style and content, both of which are image specific. For instance, in Fig. 3 (left), the style from the leopard (jaguar?) image to the content of lion image is successfully transferred, despite both having the same class (wildlife). How are style and content being disentangled then? It appears that the choice (or family) of transformations at the input of the style encoder is important in disentangling style and content. I believe a deeper discussion of this aspect is warranted. If indeed the choice of transformation is crucial, how robust would the disentangling be to the inexact/inappropriate transformation?
- Style encoding requires transformations to be hand-picked for each dataset or setting. What kind of restrictions does this impose on the way the "style" component is defined?
- I was curious about the failure cases of OverLORD, and I believe that a small set of qualitative examples would be a valuable addition. A short commentary on the limitations of the proposed approach would be very insightful.
- Competing techniques & comparisons.
How were the results for competing techniques generated? Were the trained models available? Or were they retrained by the authors? Given the general difficulties in training the large adversarial learning based models, can the authors comment on the quality of the trained models and that of the results reported? Are the competing models of StyleGAN, FUNIT, etc., trained well enough to approximately achieve results reported in the respective papers of these competing methods?

Other minor comments:
- It will be great to include the training time and GPU infrastructure used.
- In Table 1, the optima FID score is reported to be "12.9". While the classification accuracy was clear from the number of classes & random chance (also explained in text), the optimal FID score is not so obvious to me. It would be useful to add a short explanation in the text (or the caption of the table).

I was torn between the scores 5 and 6. My choice is primarily due to the lack of clarity on the role of the transformations in learning the style encoder, and the limitations or restrictions on the style factors resulting from the choice of these transformations. I am looking for deeper insights from the authors and I am open to upgrading my rating.

Post Rebuttal:
The authors have resolved my main concerns. However, the reliance on the nature of transformations applied is still an important factor. Nonetheless, I do see value in the approach presented in this paper. I am revising my rating to 6.

---

> ### Author Response · Authors · 2020-11-14
> **Response to Reviewer #5**
>
> We thank the reviewer for the dedicated review and for recognizing our “great qualitative and quantitative results”. The reviewer raised several valid points, which we believe can be easily addressed.
>
>
> **“While the results are impressive, the exposition does not justify why such an extension to LORD is a nontrivial contribution”**: This work bridges the gap between class-supervised disentanglement and image translation. Class-supervised disentanglement methods focus on disentanglement at the representation level and deal with uni-modal translations, while image translation methods aim at high-fidelity multi-modal image synthesis. To the best of our knowledge, this is the first time that image translation is performed at this level of visual quality without compromising the disentanglement of the learned representations. From a technical perspective, state-of-the-art image translation methods (e.g. StarGAN-v2, FUNIT) rely on architectural biases e.g. style injection using “AdaIN” layers which constrain the model towards local changes. As we treat all the parts of the representation in the same way within the architecture, we gain better control of their behaviour using our principled bottlenecks. We believe that this general framework would be of great interest to the image translation and disentanglement communities.
>
> **“The necessity for an adversarial loss can be perhaps better justified by an ablation”**: We added an ablation of our method without the adversarial loss in the synthesis stage to Table 1. The results confirm that high-quality disentanglement is achieved by our non-adversarial framework, while the additional adversarial loss contributes for increasing the output fidelity (FID). Qualitative evidence for this claim is presented in Fig. 11 in Appendix A.4
>
> **“However, it is not clear what is the key factor for disentangling the style and content, both of which are image specific … It appears that the choice (or family) of transformations at the input of the style encoder is important … how robust would the disentangling be to the inexact/inappropriate transformation? What kind of restrictions does this impose on the way the "style" component is defined?”**: The disentanglement of  style and content is achieved in an unsupervised manner (as we only have supervision of the class). As an inductive bias of some form is essential, the idea is to model the style as the attributes that are invariant to a specified set of transformations. While this is indeed a setting dependent design choice, we have found and shown in the paper that a set of simple operators are suitable for the common configurations in image translation in which the content should mainly encode the pose of an object (and therefore can be ignored by the style). Other settings may require more sophisticated transformations.
>
>
> **“I was curious about the failure cases of OverLORD, and I believe that a small set of qualitative examples would be a valuable addition”**: We added some qualitative examples along with a short description to Appendix A.5.
>
> **“A short commentary on the limitations of the proposed approach would be very insightful”**: We added a short discussion on the limitations of our method to Sec. 4.5.
>
> **“Competing techniques & comparisons. How were the results for competing techniques generated? Were the trained models available?”**: We used the following official publicly available pretrained models: StarGAN-v2 (for AFHQ) and StyleGAN (for mGANprior on CelebA-HQ). We have trained the rest of the baselines using the official repositories of their authors and made an effort to select the best configurations available for the target resolution (for example, FUNIT trained by us on AFHQ achieves similar results to the public StarGAN-v2 which was previously known as the SOTA on this benchmark). We added these training details to Appendix A.1.
>
> **“It will be great to include the training time and GPU infrastructure used”**: As stated in the paper, we train the disentanglement stage for 200 epochs and the synthesis stage for 100 epochs. This takes approximately ~3 days for 256x256 resolution on a single NVIDIA RTX-2080 TI. We added these training details to Appendix A.1.
>
> **“The optimal FID score is not so obvious to me”**: Following StarGAN-v2, FID between real train and test images of the same class forms the optimal score for this metric. We added this clarification to the description of the evaluation protocol (Sec. 4.3).
>
>
>
> We believe that all of the reviewer’s questions were addressed. The reviewer stated that addressing the questions would form a basis for raising the score.

---

### Decision · Program_Chairs · 2021-01-07
**Final Decision**

**Decision:**

Reject

**Comment:**

This paper was near the borderline, but ultimately, calibrating with the acceptance criteria applied to submissions across the conference, we didn't find sufficient enthusiasm among the reviewers to accept the paper.  Two reviewers put it just above the bar for acceptance, on the strength of its results.  A third reviewer finds the results to be a small improvement over other work, and finds the definitions of class, content, and style used by the authors to be confusing.  The AC agrees with the 3rd reviewer that it is more natural to define (for the class of faces) the identity to be the content and the facial pose to the the style.

Unfortunately, acceptance to ICLR required a stronger case than the reviewers presented for this paper.
The remaining concerns which swayed the AC's opinion included:

--concern that this was an incremental extension of LORD

--the reliance on the nature of transformations applied in the algorithm

--lack of any enthusiastic reviewer championing acceptance for the paper.